# Fitness and Rhizobacteria of F2, F3 Hybrids of Herbicide-Tolerant Transgenic Soybean and Wild Soybean

**DOI:** 10.3390/plants11223184

**Published:** 2022-11-21

**Authors:** Rong Liang, Xueqin Ji, Zewen Sheng, Jinyue Liu, Sheng Qiang, Xiaoling Song

**Affiliations:** Weed Research Laboratory, College of Life Sciences, Nanjing Agricultural University, Nanjing 210095, China

**Keywords:** gene flow, introgression, transgenic soybean, wild soybean, hybrids, fitness, rhizobacteria

## Abstract

The introduction of herbicide-tolerant (HT) transgenic soybeans (*Glycine max* (L.) Merr.) into farming systems raises great concern that transgenes may flow to endemic wild soybeans (*Glycine soja* Sieb. et Zucc.) via pollen, which may increase the ecological risks by increasing the fitness of hybrids under certain conditions and threaten the genetic diversity of wild soybean populations. In order to demonstrate the potential risk of gene flow from the HT soybean to the wild soybean, the fitness of F2 and F3 hybrids obtained from two wild soybean populations (HLJHRB-1, JSCZ) collected from China and the HT soybean was measured under farmland and wasteland soil conditions, as well as with or without weed competition. Compared with their wild progenitors, the F2 and F3 hybrids of HLJHRB-1 displayed a higher emergence rate, higher aboveground dry biomass, more pods and filled-seed plants, as well as better composite fitness under four planting conditions. The F2 and F3 hybrids of JSCZ also displayed a higher emergence rate, higher aboveground dry biomass, more pods, and more filled seeds per plant under mixed planting, whereas these characteristics were lower under pure planting conditions in wasteland and farmland soil. Therefore, the composite fitness of JSCZ hybrids was higher or lower depending on the planting conditions. Furthermore, the soil microbial communities of the F3 of HLJHRB-1, JSCZ, and the wild soybean were investigated with 16S rDNA sequencing, which showed that low alpha diversity of rhizobacteria was relative to high fitness, and *Rhizobium* played an important role in promoting F3 plant growth.

## 1. Introduction

Genetically modified herbicide-tolerant (HT) soybeans (*Glycine max* (L.) Merr.), individually or stacked with insect resistance, have consistently been used as a dominant trait in soybeans over the 26-year period of 1996 up until now [1,2,3]. Despite the benefits of the HT soybean, the release of HT transgenic soybeans into farming systems raises great concern that transgenes might spread to wild soybeans (*Glycine soja* Sieb. et Zucc.) via spontaneous hybridization and introgression [4,5,6,7]. This could improve the tolerance of the wild soybean to its target herbicide, create new and serious weed-control problems, and then affect agricultural production [8,9,10]. Introgression may also lead to the loss of genetic diversity in wild soybeans [11,12].

China is the center of origin and genetic diversity and geographic differentiation of the wild soybean, which is the direct ancestor of cultivated soybeans [8,13,14,15,16]. The germplasm resources of the wild soybean play a greatly important role in cultivated soybean variety improvement, such as resistance to pests [17], stress resistance [18], and yield increase [12,19]. Studies from China [4,8,20], Japan [21,22], and Korea [6,7,10] have reported gene flow between cultivated soybeans, including conventional and HT soybeans, and wild soybeans. Some medium- and large-seeded wild soybeans were proven to be hybrids of wild soybeans and cultivated soybeans [23], which is the most direct and convincing evidence confirming the natural occurrence of introgression between wild and cultivated soybeans.

Spontaneous hybridization between a transgenic crop and a compatible wild relative is only the first step for evaluating the potential gene flow risk of HT crops [5]. The fitness of the first and successive generations of hybrids determines the possibility of transgene introgression to wild populations [11,24]. Fitness is defined as the reproductive success or the proportion of genes an individual contributes to the gene pool of a population, and it is measured by vegetative and reproductive growth variables and determines whether the hybrids can survive and produce progeny [25,26]. It was confirmed that F1 hybrids produced 73–571 filled seeds per plant, although their composite fitness was significantly lower than their wild progenitors [26], and F2 hybrids demonstrated a similar fitness to wild soybean and produced at least 700 filled seeds per plant whether or not the hybrids contained the transgene [27,28].

These above studies did not pay attention to the interaction with the biotic/abiotic environment, which might influence the fitness of hybrids to a great extent. Compared with other plants, legumes usually interact more closely and abundantly with soil microorganisms, especially the rhizobia of rhizosphere bacteria, because of their symbiotic nitrogen fixation behavior [29,30,31,32,33]. Furthermore, the nitrogen fixation effect may influence the fitness of hybrids [34]. There are several research studies that have focused on the difference in plant–rhizosphere interactions between cultivated soybeans and some wild soybeans [35,36], but no study has combined the plant–rhizosphere interaction and the fitness of hybrids. An analysis of the rhizobacteria would offer another angle to evaluate the fitness of the hybrid and potential ecological risks.

The fitness of F1 under pure planting conditions in farmland soil has been studied, and it was found that two hybrids between the HT soybean and HLJHRB-1 and JSCZ wild soybean populations produced 307 and 405 average filled seeds per plant [26]. In this study, the fitness of the F2 and F3 hybrids of two wild soybean populations were evaluated under two soil conditions and with or without weed competition. Meanwhile, rhizosphere bacteria of F3 were sequenced and analyzed. The aim of the current research was to answer the issues of: (1) whether the selfed seed of F1 hybrids could germinate, grow, and reproduce in different planting conditions; (2) the effect of the planting conditions on the fitness of F2 and F3 hybrids; (3) the difference in rhizobacteria diversity and abundance between hybrids and their wild soybeans, and whether rhizobacteria is relative to the fitness of hybrids.

## 2. Results

### 2.1. Emergence Rate

The emergence rate of F2 and F3 hybrids was 82.5–94.0%, which was 10.83–13.33% higher than that of their respective wild soybeans (Figure 1).

### 2.2. Verification of Hybrids with cp4-Epsps Gene

Glyphosate-resistant gene PCR amplification tests showed that the segregation of F2 and F3 hybrids carried *cp4-epsps* gene fragments from the transgenic soybeans followed the 3:1 and 5:1 normal Mendelian segregation ratios (Table 1).

### 2.3. Size of Cotyledon and True Leaf

Compared with their wild soybeans, HLJHRB-1 F2 hybrids had a 12.06% larger cotyledon length and 30.38% larger true leaf length, whereas HLJHRB-1 F3 hybrids had only a 14.15% larger true leaf length. However, JSCZ F2 and F3 hybrids had smaller cotyledons and true leaves compared to JSCZ (Figure 2).

### 2.4. Plant Height at Third Trifoliolate Leaf Stage

Compared to their respective wild progenitors, under the same planting conditions, HLJHRB-1 F2 and F3 hybrids were 16.44–23.45 cm, which was 1.03–4.04 cm higher than HLJHRB-1. JSCZ F2 were 17.03–19.23 cm, which was similar to its wild soybean, whereas F3 was 15.89–17.56 cm, which was 1.92–2.29 cm shorter than wild JSCZ (Figure 3).

### 2.5. Aboveground Dry Biomass

HLJHRB-1 F2 and F3 hybrids weighed 49.78–166.47 g, which was 1.65–6.33 times greater than wild soybean (Figure 4A). Compared to JSCZ, JSCZ F2 and F3 hybrids were 30.59–55.64% lower under pure planting conditions but 51.51% greater or similar under mixed planting conditions (Figure 4B). The aboveground dry biomass of the weeds of HLJHRB-1 F2 and F3 hybrids was similar to that of the wild soybean, whereas the aboveground dry biomass of the weeds of JSCZ F2 and F3 hybrids was significantly lower than that of the wild soybean (Figure 4C).

### 2.6. Vitro Pollen Germination Rate

Compared to its wild soybean, the pollen germination rate of HLJHRB-1 F2 was similar or 15.41–16.87% greater in pure and mixed planting conditions, respectively. The germination rate of HLJHRB-1 F3 hybrids was approximately 7% lower under the four planting conditions. JSCZ F2 under the four planting conditions and F3 under pure planting conditions were 8.98–12.45% lower than the wild soybean. Meanwhile, JSCZ F3 was 6.91% greater and similar under mixed planting conditions in wasteland and farmland soil, respectively. In general, both hybrids and wild soybeans had a greater pollen germination rate under pure conditions compared to mixed planting conditions (Figure 5).

### 2.7. Pod Number and Filled-Seed Number Per Plant

HLJHRB-1 F2 and F3 hybrids produced 343–1113 pods and 533–2101 filled seeds per plant, which was 2.77–6.46 times and 2.44–6.26 times more than their wild soybeans under the same planting conditions (Figure 6A,C). JSCZ F2 and F3 hybrids produced 343–922 pods and 743–1698 filled seeds per plant under pure planting conditions, which was 13.87–43.53% and 19.22–46.59% less than their wild soybeans, and 279–632 pods and 470–1439 filled seeds per plant under mixed planting conditions, which was 24.28–102.78% and 28.70–114.98% more than their wild soybeans (Figure 6B,D). Overall, the F2 and F3 hybrids of HLJHRB-1 and JSCZ displayed different tendencies with their wild soybeans under pure and mixed planting conditions.

### 2.8. 100-Seed Weight

The majority of F2 and F3 hybrids were 1.06–1.58 g, which was 0.92–34.89% lower than their wild soybean under four planting conditions, whereas the HLJHRB-1 F3 was similar to its wild soybean (Figure 7). The planting conditions did not significantly affect the 100-seed weight of F2 and F3 hybrids.

### 2.9. Relative Composite Fitness

Under the same planting conditions, the composite fitness of HLJHRB-1 F2 and F3 was significantly higher than the wild soybean. Compared to JSCZ, the composite fitness of JSCZ F2 and F3 was significantly lower or similar under pure and mixed planting conditions, respectively (Figure 8).

### 2.10. HLJHRB-1 F3 Rhizobacteria under Pure Planting Conditions in Farmland Soil

The coverage rate of each sample was more than 96%, which reflects the real situation of the test samples. The estimated richness of Chao1, the Shannon index, and the evenness index Shannoneven of the HLJHRB-1 F3 were significantly lower than its wild soybean (Table 2). That is, the rhizosphere bacterial richness, diversity, and distribution evenness of HLJHRB-1 F3 were the lowest among CK, HLJHRB-1 F3, and HLJHRB-1.

The distance between each sample of HLJHRB-1 F3 CK and HLJHRB-1 is shown in Figure 9A. CK effectively separated from soil which planted HLJHRB-1 and HLJHRB-1 F3 at the PC2 level, with an explanation degree of 24%, whereas HLJHRB-1 and HLJHRB-1 F3 separated from each other at the PC1 level, with an explanation degree of 38% (Figure 9B).

### 2.11. JSCZ F3 Rhizobacteria

The coverage rate of each sample was more than 96.5%, which reflects the real situation of the samples. The Chao1 value of JSCZ F3 and its wild soybean had no significant difference, which means the richness of the rhizobacteria of JSCZ F3 and its wild soybean were similar. The Shannoneven index of JSCZ F3 and its wild soybean was around 0.8, which means the different species in the rhizosphere were well distributed. However, the evenness of the rhizobacteria of JSCZ F3 under pure planting conditions was significantly higher than JSCZ, and the evenness of rhizobacteria of JSCZ F3 under the four planting conditions did not differ significantly (Table 3). The Shannon index of JSCZ F3, which indicates the diversity of its rhizobacteria, was significantly higher under mixed planting conditions in wasteland soil and lower under pure planting conditions in farmland soil.

At the phylum level, the rhizobacteria of JSCZ and JSCZ F3 was significantly different under pure planting conditions in farmland soil (Figure 10A). For JSCZ F3, the influence of competition is greater than that of soil. At the Specie (OTU) level, it was shown that the differences of the rhizobacteria of JSCZ and F3 under mixed planting conditions were less than those of pure planting (Figure 10B). Only JSCZ F3 in farmland soil was separated effectively at the phylum level (Figure 10C), and the two principal components together could explain 80% of the difference. At the species level (Figure 10D), PC2 separated the control wasteland soil from the farmland soil, but the farmland soil of the JSCZ and F3 overlapped, and the JSCZ F3 of different soils overlapped to another group.

In a comparison among the four planting conditions, the relative abundance of *Rhizobiaceae* and *Rhizobium* of JSCZ F3 under pure planting conditions in farmland soil was significantly higher than that under pure planting conditions in wasteland soil and mixed planting conditions in farmland soil, respectively (Figure 11A). Comparing the same planting conditions, the relative abundance of the *Rhizobium* of JSCZ F3 was significantly higher than that of JSCZ under pure and mixed planting conditions in wasteland soil (Figure 11B).

## 3. Discussion

### 3.1. Germination, Cotyledon, and True Leave Size

The growth cycle of annual plants, including legumes, depends on seed germination [37,38]. In the current research, all F2 and F3 hybrids emerged at 72.67–94.00%, which was higher or similar compared to those of their wild progenitors. The explanation should be a higher or similar seed viability rather than seed coat impermeability of the wild soybean, because the seed coat of hybrids and wild soybeans was broken with the same method. The seed viability of F2 and F3 increased significantly more than F1, which had a lower emergence rate than their wild soybeans [26]. Although cultivated soybeans and wild soybeans carry similar genomes (GG, 2n = 40), meiotic aberrations and heteromorphic chromosome pairing were observed in hybrids [39]. The chromosomes in F2 and F3 should be more stable than F1 due to the loss of unstable paired chromosomes in meiosis as the generation increases [40,41].

Cotyledon size determines the initial nutrient supply during seed germination, and the first pair of true leaves determines the photosynthesis ability of seedlings after germination, which is the basis for seedling colonization and has a great impact on the competitiveness of plants in the subsequent growth process [42,43,44]. The cotyledons and true leaves of the two F2 and F3 hybrids were smaller, especially the JSCZ hybrids, although the true leaf length of some hybrids was longer than their wild soybean. The smaller cotyledon and true leaf size may not be a benefit for seedling colonization. In a previous study, these two F1 hybrids emerged at 30.85 and 65.97%, which was lower than that of their wild progenitors, and two kinds of F1 hybrids were significantly smaller in both length and width, as were the true leaves [26]. The above results indicate that these hybrids significantly increased their seed viability, but seedling colonization was not improved after the self-pollination of F1 hybrids. A higher emergence rate may be a beneficial factor for hybrids establishing populations. Although more filled seeds of the hybrids could emerge, the seedling viability of the hybrids might be weaker than that of its corresponding wild soybean.

### 3.2. Fitness and Rhizobacteria of HLJHRB-1 F2 and F3 Hybrid

The plant height, aboveground dry biomass, pods, and filled seeds per plant of HLJHRB-1 F2 and F3 displayed similar tendencies, being significantly greater than their wild soybeans under the four planting conditions. This may be due to the different adaptations of soybean populations to the climate, especially the photoperiod [45]. A lower latitude for plants from a high latitude means a shorter photoperiod, higher temperature, higher moisture, a reduced assimilation rate, and a shorter reproductive period [46,47,48,49]. The maternal parent HLJHRB-1 suffered from unfit condition in the experimental area that is optimal for the paternal parent HT soybean. Influenced by the genome from the HT soybean, HLJHRB-1 F2 and F3 had a greater fitness than their wild soybeans.

However, the 100-seed weight of HLJHRB-1 F2 and F3 hybrids was under 2.0 g, which should be classified as small-seeded according to the genetic categories of the Chinese wild soybean, whereas the 100-seed weight of HLJHRB-1 was 1.89–2.32 g and 1.12–1.17 g in two years, some of which should be medium-seeded [8]. In the current research, the 100-seed weight of HLJHRB-1 F2 and F3 hybrids, as well as HLJHRB-1, was less affected by planting conditions. A single seed with more nutrients will help the plant to improve its competitiveness in a complex environment [50]. Although each seed of HLJHRB-1 F2 and F3 was less than their wild soybeans, they still produced more filled seeds. Wild soybeans from a high latitude such as HLJHRB-1 rarely hybridizing naturally with cultivated soybeans in low latitudes to produce hybrids with high fitness. A potential risk would occur if wild soybean of high latitude was mixed with thee cultivated soybean and transported to low altitude area. Therefore, it is necessary to raise quarantine standards for cultivated soybean from high latitude.

The rhizosphere has profound impacts on plant growth and health, and plants also play essential roles in specializing metabolites in shaping the rhizosphere microbiome [51,52,53,54,55,56]. In the pure-planting farmland soil, HLJHRB-1 F3 performed better in terms of its aboveground dry biomass and reproductive index and had a significantly lower alpha diversity in rhizosphere bacteria. This indicates that HLJHRB-1 F3 plants could screen rhizosphere bacteria [53,57,58]. Therefore, we speculate that HLJHRB-1 F3 had a better ability than wild soybean to amplify beneficial bacteria, including rhizobia, and to inhibit harmful bacteria. This ability of HLJHRB-1 F3 may be inherited from its paternal plant, the HT soybean. The transgenes or other cultivated soybean genes that confer an advantage may be included in the HLJHRB-1 F2 and F3 hybrids. Therefore, this selection may be not caused by the HT gene, because there was no comparative experiment with HT negative hybrids. As such, the relationship between rhizobacteria and the fitness of HLJHRB-1 F3 needs to be further studied.

### 3.3. Fitness and Rhizobacteria of JSCZ Hybrid

The JSCZ F2 and F3 and the wild soybean obviously performed differently under different planting conditions. JSCZ F2 and F3 demonstrated a greater advantage in competition with weeds than wild soybeans, especially in wasteland soil. From another angle, the aboveground dry biomass of the weeds in JSCZ F2 and F3 was significantly less than that in wild soybean in both farmland and wasteland soil. This shows that JSCZ F2 and F3 were more competitive with weeds than wild soybeans. The reason may be due to the interaction among JSCZ F2 and F3, weeds, and their rhizobacteria. The interaction among the abiotic environment, plants, and microbial communities is a complex, integrated system [29,59,60,61]. As shown by the results of the PCA, the rhizobacteria of JSCZ and JSCZ F3 under mixed planting conditions in farmland soil effectively separated. Furthermore, the rhizobacteria of JSCZ F3 under mixed planting conditions in wasteland soil was at the same PC1 level as that of JSCZ under pure planting conditions in both kind of soils, and at the same PC2 level as that of farmland soil CK. This indicates that JSCZ F3 changed the diversity and abundance of rhizobacteria under mixed planting conditions in wasteland soil and finally increased its competitive ability. Simulations of mathematical models demonstrated that the assembly of microbial communities by plants effects their competition with other plants [62].

*Rhizobia* plays an important role in biological nitrogen fixation as well as nutrition uptake [58,60,63,64,65,66]. The post hoc test showed an association between the abundance of *Rhizobiaceae* and *Rhizobium* in rhizobacteria and the fitness of JSCZ F3. A higher abundance of *Rhizobiaceae* and *Rhizobium* under pure planting conditions in farmland soil was consistent with higher aboveground dry biomass, more pods, and more filled seeds per plant versus mixed planting conditions in farmland soil and under pure planting conditions in wasteland soil. In wasteland soil, the abundance of *Rhizobium* of JSCZ F3 was significantly higher than its wild soybean, which was also consistent with the difference shown in fitness. In general, differences in rhizobacteria partly explain the fitness of JSCZ F3 and its wild soybean.

## 4. Materials and Methods

Herbicide-tolerant transgenic soybeans (T14R1251–70) were provided by the National Soybean Improvement Center of Nanjing Agricultural University. The HT soybean, containing one single-copy *cp4-epsps*, was obtained by *Agrobacterium*-mediated cotransformation of the receptor soybean NJR44-1, which is an elite line bred by the National Soybean Improvement Center of Nanjing Agricultural University. The HT soybean withstands 3600 g a.i. ha-1 41% glyphosate isopropylammonium AS (Roundup Ultra; Monsanto, St. Louis, MO, USA). Wild soybean populations were collected from Heilongjiang and Jiangsu (Table 4), then named after the province and the city. Crossed seeds were obtained by artificial hybridization of wild soybeans as the maternal plants and HT soybeans as the paternal plants from 2016 to 2017 [5]. The crossed seeds were harvested from different mother plants, mixed, and then stored at 4 °C until further use. Experiments were conducted in a greenhouse and net house at the Pailou Experimental Farm (32°20′ N, 118°37′ E), Nanjing Agricultural University, China, from 2018 to 2020 (Table 5).

### 4.1. Seed Sowing and Emergence

At the beginning of May of the planting year, 120 filled seeds of each wild soybean population and hybrid seeds were selected from the 10 mother plants (8–15 seeds from each plant). Seed coats of wild soybeans and the hybrids were sturdy and durable under their natural state, so the embryo-dorsal seed coats (on the opposite of the hilum) of the wild soybeans and crossed seeds were carefully nicked with a razor blade prior to sowing (the seed coat was broken, but the internal structure of the seeds was undamaged) to break the limit of imperviousness of the seed coat. Then, a single filled seed of each wild soybean population and their hybrids were sown at 1 cm depth in individual pots (7 cm diameter, 7.5 cm height) previously filled with a mixture of wasteland soil and organic cultivated soil (Green Island Horticultural Development Center, Zhenjiang, China) at a 1:1 (*v*/*v*) ratio. Pots were laid out in a completely randomized design in the same replicate in the greenhouse. All emerging hybrid seedlings were tested by PCR to confirm whether they contained the *cp4-epsps* gene.

### 4.2. Seedling Transplanting and Variables Measured

#### 4.2.1. Without Weed Competition

A total of 15 or 20 uniform plants in size of wild soybeans and hybrids with the *cp4-epsps* gene were transplanted individually into pots with holes at the bottom (23 diameter cm, 25 cm height) containing the farmland soil and wasteland soil, as described in Table 6, when the second trifoliolate leaf spread completely. On the third day after transplanting, a bamboo pole (2 diameter cm, 200 cm height) was inserted into the pots with wild soybeans and hybrid seedlings for the plants to climb. Pots were watered and hand-weeded as needed. No chemicals were applied during the experiment. Seedlings were grown under natural conditions exposed to natural light (approximately 11–14 h/day) and temperature (approximately 15–35 °C) from the date of transplanting to harvesting (from the end of June to the end of November). Adjacent pots were separated by 60 cm. Pots were laid out in a completely randomized design in the net house, and no sexually compatible Leguminosae species were present for a 100 m radius around the experiment. The plant height was measured from the top of the plant to the cotyledonary node when the third trifoliolate leaf spread completely. The other fitness components were measured as follows. Pollen viability was tested at the full flowering stage. Pollen was collected from nascent flowers at 7–8 a.m., and the in vitro pollen germination rate at 60 min was tested according to the method described by Liu [26]. At least 50 pollen grains from five flower buds on each of the one to three plants for the wild soybean and hybrids were used as one replicate, and a total of nine replicates were assessed each time. Finally, the in vitro pollen germination rate was calculated as follows: (pollen germinated/pollen observed) ×100. When the pollen tube length was twice the pollen grain length, it was considered to have germinated. When 100% of pods darkened (harvest maturity), each individual plant was separately harvested (cut from cotyledonary node). Each plant was sun dried to a constant weight, and the aboveground dry biomass was weighed. The number of pods of each harvested plant was counted. All seeds were hand-peeled from the pods. Then, the number of filled seeds was counted for each plant. After being sun dried for 10 days in a greenhouse, 100 filled seeds were randomly counted from 10 plants and weighed for each wild soybean, transgenic soybean, and F1 hybrid.

#### 4.2.2. With Weed Competition

On the same day of sowing the wild soybeans, hybrids in the experiment without weed competition, 0.5 g of seeds each from *Setaria viridis* (L.) Beauv., *Digitaria sanguinalis* (L.) Scop., *Echinochloa colona* (L.) Link., and 0.25 g of seeds from *Eleusine indica* (L.) Gaertn. were well mixed and then sown evenly on the surface of the pots with holes in the bottom (52 cm diameter, 35 cm height). The pots contained the same media as those in the experiment without competition. All the methods for measuring performance variables, including plant height, aboveground dry biomass, pod number per plant, and filled seed number per plant, were the same as those used in the experiment without weed competition.

### 4.3. Procedures to Verify Hybrids with cp4-Epsps Gene

All emerging seedlings were tested by PCR to confirm whether they contained the *cp4-epsps* gene. The procedure was the same as that described by Liu [26]. The number of plants carrying the *cp4-epsps* gene were counted. Then, whether or not the segregation of plants carrying the *cp4-epsps* gene followed the 3:1 and 5:1 ratio, normal Mendelian segregation ratios were analyzed as follows:(1)$$\chi^{2}=\frac{{\left[\left|b\times A_{1}-a\times A_{2}\right|-(a+b)/2\right]}^{2}}{a\times (A_{1}+A_{2})}$$
when $\chi^{2}$ < 3.84, *p* > 0.05, this means the *cp4-epsps* gene was transferred to hybrids and followed normal Mendelian segregation ratios. *A*_1_ indicates the number of plants carrying the *cp4-epsps* gene. *A*_2_ is the number of plants that did not carry the *cp4-epsps* gene. For F2 hybrids, *a* = 3, *b* = 1; for F3 hybrids, *a* = 5, *b* = 1.

### 4.4. Rhizosphere Soil Sampling and 16s rDNA High-Throughput Sequencing

When the plants were harvested, the soil was taken out adhering to the plant roots, sieved, and well mixed. The plant root debris were removed, and the soil was collected and store at −80 °C for later use. The primers used for PCR amplification were V4 region primers in the bacterial 16S rRNA gene (515F: GTGCCAGCMGCCGCGGTAA, 806R: GGACTACHVGGGTWTCTAAT). High-throughput analysis of the soil nitrogen-fixing microbial community structure was carried out using the Illumina Miseq^TM^ sequencing platform. After sample collection, sample processing, high-throughput sequencing, and preliminary data processing were entrusted to Sangon Biotech (Shanghai) Co., Ltd. The sequencing data were preprocessed and annotated, and species information was counted with Qiime2. The alpha diversity analysis was conducted with mothur. The sample distance matrix was obtained with phyloseq, PCA analysis was carried out with R, ANOVA analysis and the post hoc test were conducted with STAMP, and the data were plotted with Prism GraphPad.

## 5. Conclusions

In general, the fitness of HLJHRB-1 F2 and F3 whose maternal parent is from high latitudes is higher than their wild soybeans. This may be caused by the paternal parent HT soybean. Compared to JSCZ, the fitness of JSCZ F2 and F3 was more complicated depending on the planting conditions. These differences may be partially explained by the interaction among hybrids or soybeans, weeds, and rhizobacteria. The results imply that the F2 and F3 hybrids may have a high ecological risk; therefore, the cultivation of HT soybeans in open space should be allowed under strict control in East Asia where the wild soybean is found.

## Figures and Tables

**Figure 1 plants-11-03184-f001:**
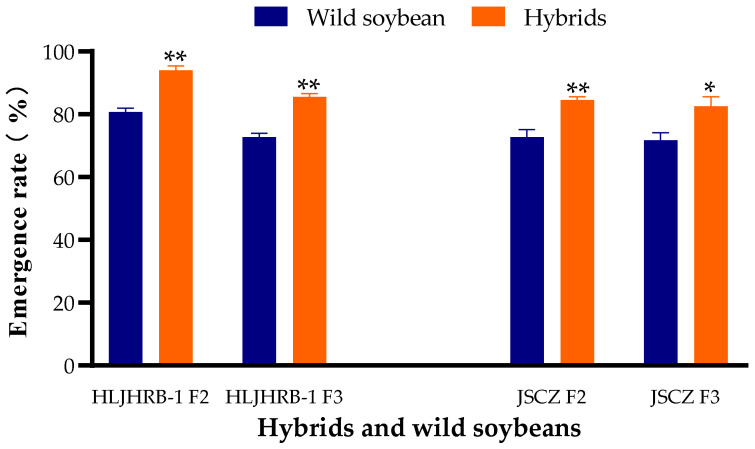
Emergence rate of F2, F3 hybrids and their wild soybeans. Note: * and ** indicate significant difference (*p* < 0.05) and extremely significant difference (*p* < 0.01) between hybrids and their wild soybeans.

**Figure 2 plants-11-03184-f002:**
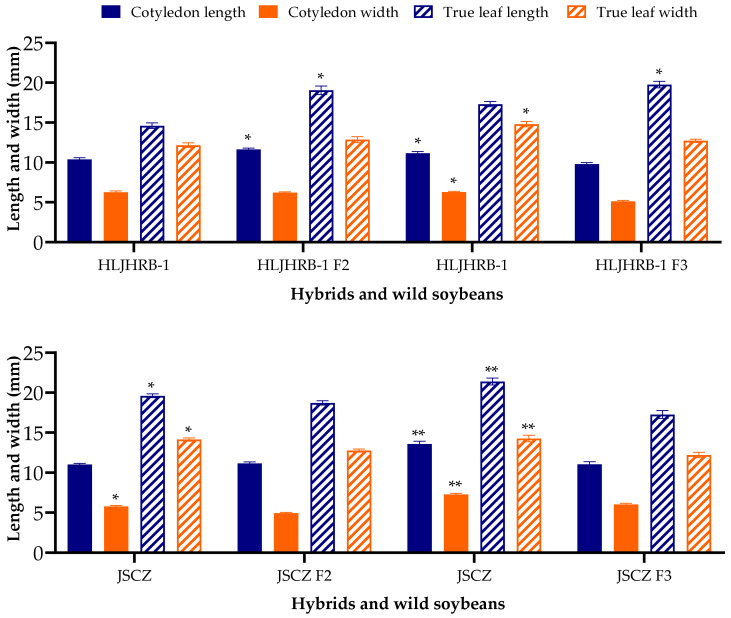
Size of cotyledon and true leaves of hybrids and wild soybeans. Note: * and ** indicate significant difference (*p* < 0.05) and extremely significant difference (*p* < 0.01) of the same trait between hybrids and their wild soybeans.

**Figure 3 plants-11-03184-f003:**
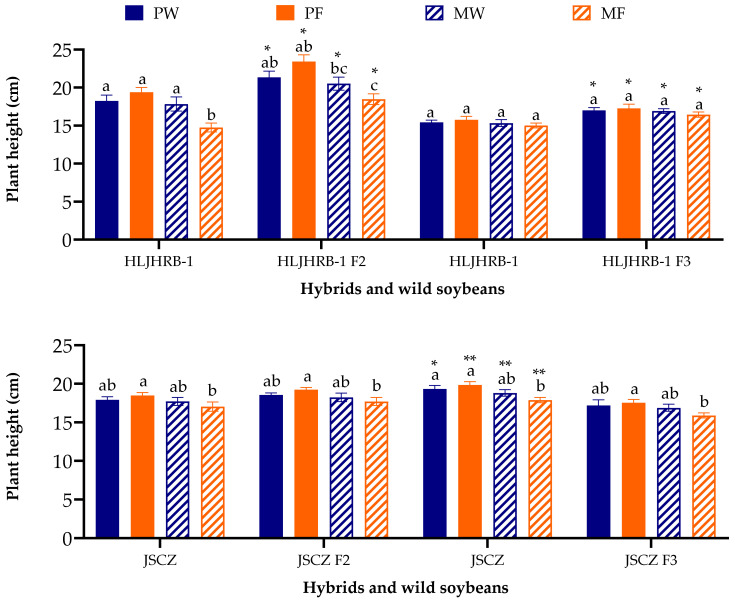
Plant height of hybrids and wild soybean under four planting conditions (the third trifoliolate leaf stage). Note: * and ** indicate significant difference (*p* < 0.05) and extremely significant difference (*p* < 0.01) between hybrids and their wild soybeans. Different lowercase letters indicate significant difference (*p* < 0.05) of hybrids or wild soybeans among four planting conditions.

**Figure 4 plants-11-03184-f004:**
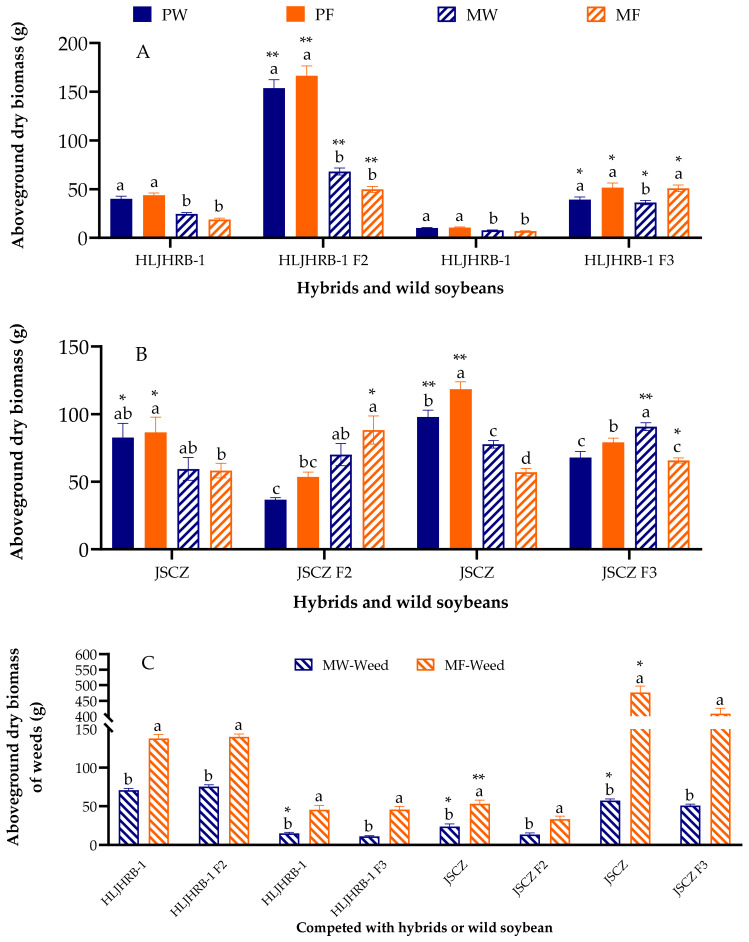
Aboveground dry biomass of hybrids, wild soybeans (**A**,**B**), and weeds (**C**) under four planting conditions. Note: * and ** indicate significant difference (*p* < 0.05) and extremely significant difference (*p* < 0.01) of the same trait between hybrids and their wild soybeans. Different lowercase letters indicate significant difference (*p* < 0.05) of hybrids or wild soybeans among four planting conditions.

**Figure 5 plants-11-03184-f005:**
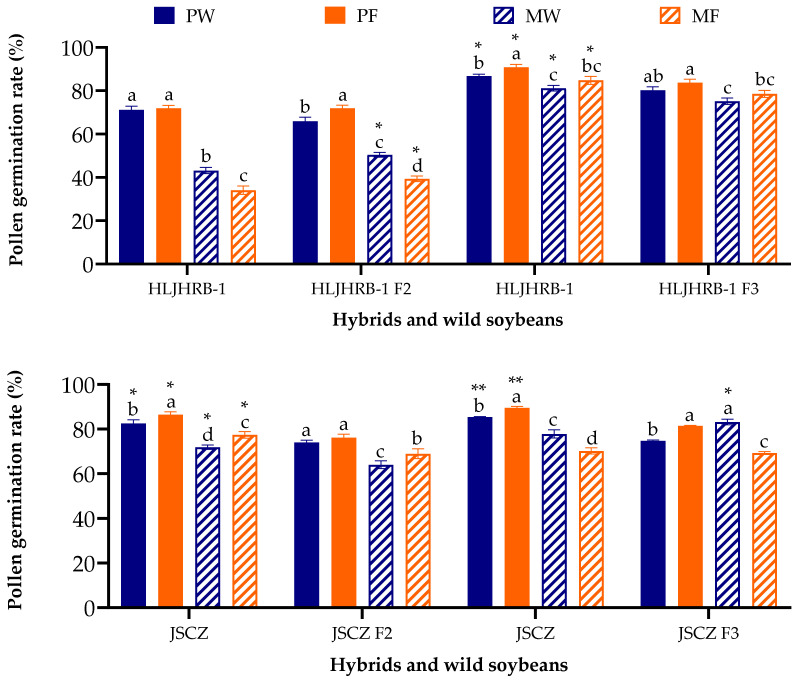
Vitro pollen germination rate of hybrids and wild soybeans at 60 min under four planting conditions. Note: * and ** indicate significant difference (*p* < 0.05) and extremely significant difference (*p* < 0.01) of the same trait between hybrids and their wild soybeans. Different lowercase letters indicate significant difference (*p* < 0.05) of hybrids or wild soybeans among four planting conditions.

**Figure 6 plants-11-03184-f006:**
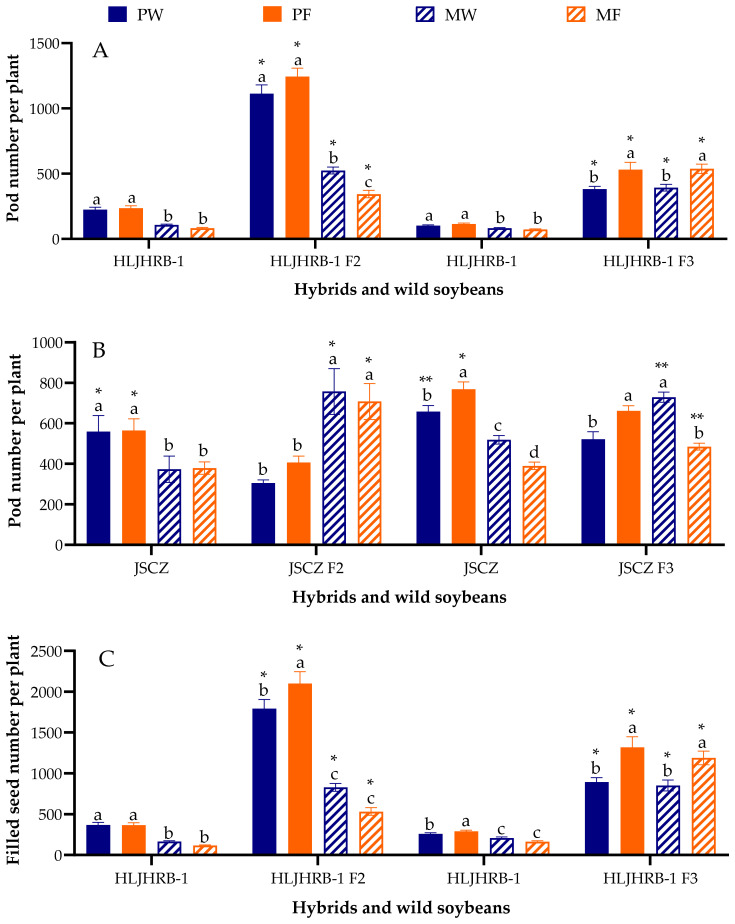
Pod number (**A**,**B**) and filled-seed number per plant (**C**,**D**) of hybrids and wild soybeans under four planting condition. Note: * and ** indicate significant difference (*p* < 0.05) and extremely significant difference (*p* < 0.01) of the same trait between hybrids and their wild soybeans. Different lowercase letters indicate significant difference (*p* < 0.05) of hybrids or wild soybeans among four planting conditions.

**Figure 7 plants-11-03184-f007:**
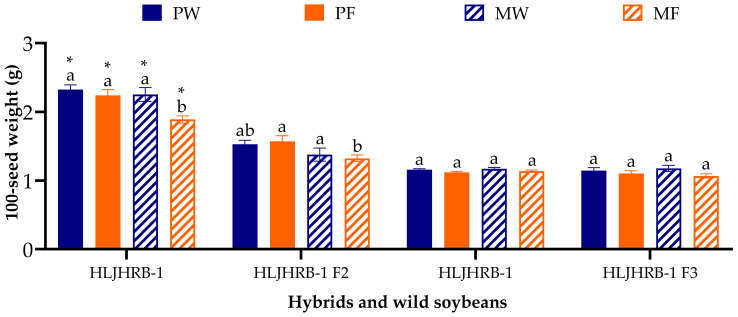
The 100-seed weight of hybrids’ seeds and wild soybean seeds under four planting conditions. Note: * and ** indicate significant difference (*p* < 0.05) and extremely significant difference (*p* < 0.01) of the same trait between hybrids and their wild soybeans. Different lowercase letters indicate significant difference (*p* < 0.05) of hybrids or wild soybeans among four planting conditions.

**Figure 8 plants-11-03184-f008:**
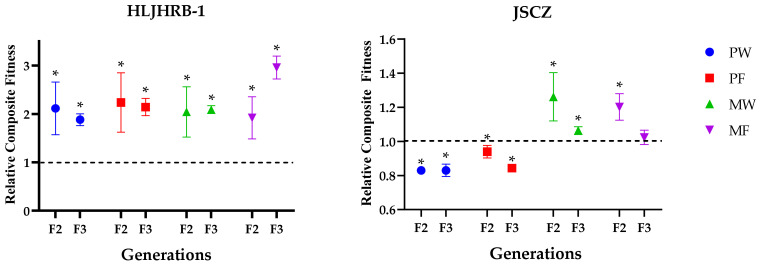
Comparison of composite fitness between wild soybeans and hybrids. Note: the dashed line represents the composite fitness of wild soybean as 1; * indicates significant difference (*p* < 0.05) of the same trait between hybrids and the wild soybeans.

**Figure 9 plants-11-03184-f009:**
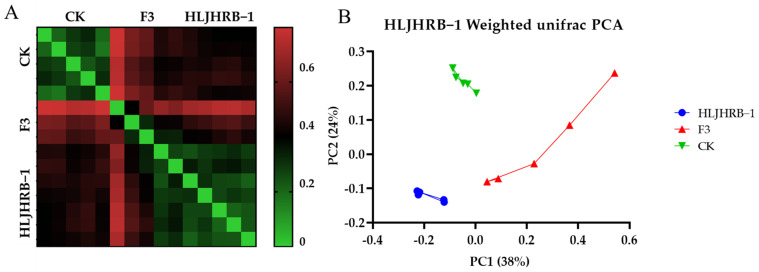
Distance for heatmap (**A**) and PCA (**B**) of HLJHRB-1 and F3.

**Figure 10 plants-11-03184-f010:**
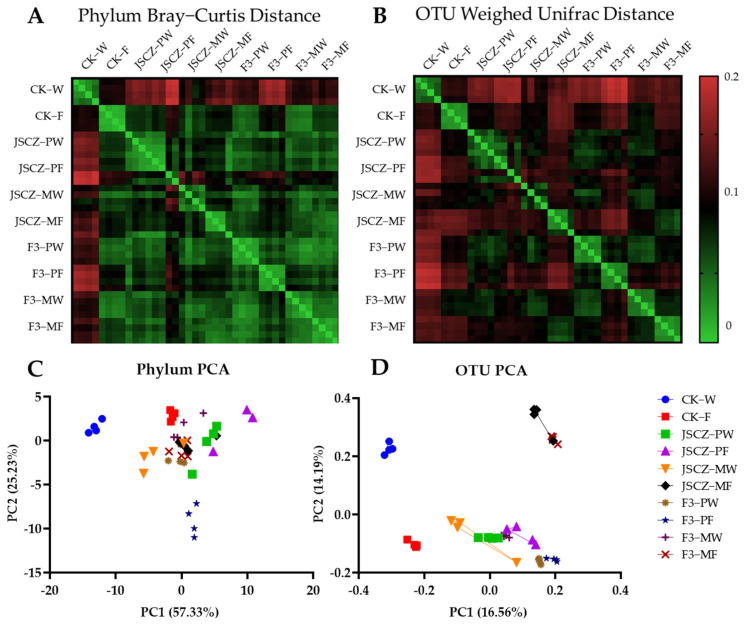
Distance for heatmap (**A**,**B**) and PCA (**C**,**D**) in different classification levels of JSCZ and F3.

**Figure 11 plants-11-03184-f011:**
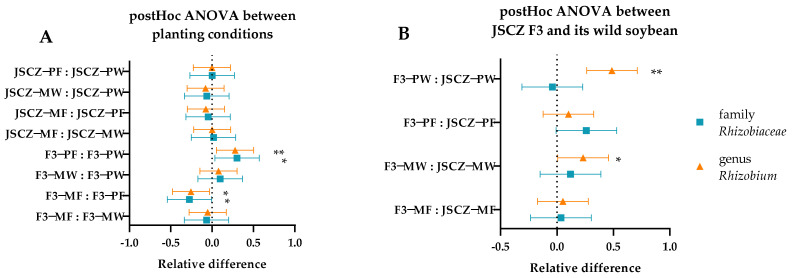
Post hoc of *Rhizobiaceae* and *Rhizobium* between JSCZ and F3. Note: (**A**) * and ** indicate significant difference (*p* < 0.05) and extremely significant difference (*p* < 0.01) of JSCZ F3 among the four planting conditions; (**B**) * and ** indicate significant difference (*p* < 0.05) and extremely significant difference (*p* < 0.01) between JSCZ F3 and its wild soybean. The positive and negative values of relative abundance reflect the relationship between the former and the latter.

**Table 1 plants-11-03184-t001:** Analysis of hybrids carrying *cp4-epsps* gene and χ^2^ test of F2, F3 hybrids.

Hybrids	PNW	PNWO	TSR	χ^2^	*p*
HLJHRB-1 F2	216	84	3:1	1.28	>0.05
JSCZ F2	105	45	3:1	1.74	>0.05
HLJHRB-1 F3	241	59	5:1	1.73	>0.05
JSCZ F3	119	31	5:1	1.45	>0.05

Note: χ^2^ < 3.84, *p* > 0.05, indicating that *cp4-epsps* gene was transferred to hybrids and followed normal Mendelian segregation ratios. PNW means plant number with *cp4-epsps* gene; PNWO means plant number without *cp4-epsps* gene; TSR means theoretical segregation ratios.

**Table 2 plants-11-03184-t002:** Alpha diversity of rhizosphere soil of CK, HLJHRB-1 F3, and HLJHRB-1.

Soybeans	Coverage	Chao1	Shannoneven	Shannon
CK	0.98	5144.64 ± 275.4 a	0.64 ± 0.02 a	5.4 ± 0.22 a
HLJHRB-1 F3	0.97	3786.07 ± 401.95 b	0.43 ± 0.05 c	3.42 ± 0.4 c
HLJHRB-1	0.97	5175.57 ± 205.72 a	0.56 ± 0.02 ab	4.61 ± 0.17 ab

Note: Different lowercase letters indicate significant difference (*p* < 0.05) among CK, HLJHRB-1 F3, and HLJHRB-1.

**Table 3 plants-11-03184-t003:** Alpha diversity of rhizosphere soil of CK, JSCZ, and JSCZ F3.

Planting Conditions	Soybeans	Coverage (%)	Richness Index	Evenness Index	Diversity Index
			Chao1	Shannoneven	Shannon
PW	CK-W	97.48%	8652.69 ± 534.61 a	0.79 ± 0.00 a	6.88 ± 0.08 a
	JSCZ	97.84%	8451.91 ± 225.01 Ba	0.76 ± 0.00 Cb	6.62 ± 0.02 Cb
	JSCZ F3	97.66%	8562.90 ± 155.92 Aa	0.79 ± 0.00 Aa	6.87 ± 0.03 Ba
PF	CK-F	97.30%	9057.88 ± 175.51 a	0.80 ± 0.00 a	6.98 ± 0.03 a
	JSCZ	98.25%	7764.78 ± 146.50 Cab	0.76 ± 0.00 Cc	6.56 ± 0.04 Cc
	JSCZ F3	97.91%	7043.98 ± 832.06 Bb	0.79 ± 0.01 Ab	6.68 ± 0.02 Cb
MW	CK-W	97.48%	8652.69 ± 534.61 b	0.79 ± 0.00 b	6.88 ± 0.08 b
	JSCZ	96.91%	9462.12 ± 256.79 Aab	0.81 ± 0.01 Aa	7.13 ± 0.06 Aa
	JSCZ F3	97.71%	9891.08 ± 236.07 Aa	0.80 ± 0.01 Aab	7.07 ± 0.02 Aa
MF	CK-F	97.30%	9057.88 ± 175.51 a	0.80 ± 0.00 a	6.98 ± 0.03 a
	JSCZ	97.54%	8784.09 ± 214.60 Ba	0.79 ± 0.00 Ba	6.92 ± 0.03 Ba
	JSCZ F3	97.83%	8754.58 ± 229.05 Aa	0.79 ± 0.00 Aa	6.89 ± 0.01 Ba

Note: Different uppercase letters indicate significant difference (*p* < 0.05) of hybrids or wild soybeans among four planting conditions; different lowercase letters indicate significant difference (*p* < 0.05) among CK, hybrids and their wild soybeans at the same planting condition.

**Table 4 plants-11-03184-t004:** Information of wild soybeans used in the experiment.

Population	Collecting Site	Latitude and Longitude
HLJHRB-1	Harbin City, Heilongjiang Province	N46°06′34″, E127°21′43″
JSCZ	Changzhou City, Jiangsu Province	N31°37′13″, E119°29′53″

**Table 5 plants-11-03184-t005:** Seed collection and planting time of tested hybrids.

Hybrids	Planting Years	Hybrids	Planting Years
HLJHRB-1 F2	2018	HLJHRB-1 F3	2019
JSCZ F2	2019	JSCZ F3	2020

**Table 6 plants-11-03184-t006:** Soil physicochemical properties per year.

Year	Soil Conditions	Organic Matter g/kg	Total Nitrogen g/kg	Total Phosphorus g/kg	Total Potassium g/kg	Available Phosphorus mg/kg	Alkali-Hydrolyzable Nitrogen mg/kg
2018	Wasteland	2.79	0.37	0.56	22.04	22.39	44.15
2018	Farmland	38.51	2.2	1.76	18.94	47.81	163.74
2019	Wasteland	4.82	0.27	0.17	9.79	0.10	10.71
2019	Farmland	9.74	0.37	0.26	10.07	1.68	23.59
2020	Wasteland	7.78	0.72	0.25	20.94	9.99	51.91
2020	Farmland	11.19	1.06	0.36	21.09	28.21	145.41

## Data Availability

The data presented in this study are available in the article.

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
