# Peer review of "Fitness and Rhizobacteria of F2, F3 Hybrids of Herbicide-Tolerant Transgenic Soybean and Wild Soybean"

_plants, 2022, doi:10.3390/plants11223184_

Round 1

Author Response

Dear reviewer,

Thank you for your comments of the article. For your suggestion of abbreviation section, we added an annotation after the full expressions when an abbreviation firstly appeared.

We hope these modifications could be formal and normative enough to be published, and we are really appreciated for your advises.

Yours sincerely,

Reviewer 2 Report

Dear Editor

The article entitled 'Fitness and rhizobacteria of F2, F3 hybrids of herbicide tolerant transgenic soybean and wild soybean' is a useful finding and may be published after a minor revision. My comments for the article is below:

1. What do the authors mean by the word 'fitness' in the title? I think, all words in the title should be clear

2. It should be great if the authors could add 1-2 sentences in the Abstract initially for the importance of the study.

3. I found some irrelevant information in the Introduction section which must be rephrased

4. M & Sections should be clear, particularly methods used for the observation

5. Results of the study should be based on data in all Figures and Tables. Also Tables and Figures citations should be replaced in the appropriate place

Finally I recommended to publish it with a minor revision.

Author Response

Dear reviewer,

Thank you for your comments on the article entitled “Fitness and rhizobacteria of F2, F3 hybrids of herbicide tolerant transgenic soybean and wild soybean”. We’ll response your suggestions as follow:

  1. What do the authors mean by the word 'fitness' in the title? I think, all words in the title should be clear

Fitness was defined as the relative ability of an individual to survive and successfully reproduce in a given environment (Jenczewski et al. 2003). It is broadly used in the research of plant ecological risk, such as “Engineered Genes in Wild Populations: Fitness of Weed-Crop Hybrids of Raphanus Sativus (Ecological Applications Volume 4, Issue 1 p. 117-120, 1994)” and “Root-secreted bitter triterpene modulates the rhizosphere microbiota to improve plant fitness (Nature Plants Volume 8, p. 887–896, 2022)”. We still think that “fitness” is propriate and clear enough to describe the valuation of the ability of the hybrids of herbicide tolerant transgenic soybean and wild soybean.

  1. It should be great if the authors could add 1-2 sentences in the Abstract initially for the importance of the study.

Thank you for your suggestion. We added “The introduction of herbicide tolerant (HT) transgenic soybean [Glycine max (L.) Merr.] into farming systems raises great concern that transgenes may flow to endemic wild soybean (Glycine soja Sieb. et Zucc.) via pollen, which may increase the ecological risks by increasing the fitness of hybrids under certain conditions and threaten the genetic diversity of wild soybean populations” at the beginning of the Abstract.

  1. I found some irrelevant information in the Introduction section which must be rephrased

Thank you for your suggestion. We checked every paragraph of Introduction and deleted the first sentence of the advantages of transgenic soybean. It may be more relevant to the topic of the article to begin with the widely application of herbicide tolerant soybeans and lead to the necessity of the study of gene flow and hybrids.

  1. M & Sections should be clear, particularly methods used for the observation

Thank you for your reminding. We added the observation methods in 4.2.1 and 4.2.2, which are clear enough to describe all the data measurement in Results.

  1. Results of the study should be based on data in all Figures and Tables. Also Tables and Figures citations should be replaced in the appropriate place

Thank you for your reminding. We checked all the citations of tables and figures, replaced the citations in 2.6, 2.7, 2.11 and 4 to confirm that they were in an appropriate place.

Finally, we appreciate your comments and advises for the article to make it more appropriate and formal to be published.

Yours sincerely